# Pharmacovigilance in High-Income Countries: Current Developments and a Review of Literature

**DOI:** 10.3390/pharmacy11010010

**Published:** 2023-01-06

**Authors:** Muhammad Akhtar Abbas Khan, Saima Hamid, Zaheer-Ud-Din Babar

**Affiliations:** 1Health Services Academy Islamabad, Chak Shahzad, Islamabad 44000, Pakistan; 2Fatima Jinnah Women University, Rawalpindi 46000, Pakistan; 3Center for Pharmaceutical Policy and Practice Research, Department of Pharmacy, School of Applied Sciences, University of Huddersfield, Huddersfield HD1 3DH, UK

**Keywords:** pharmacovigilance, adverse drug reactions, high-income countries, literature review

## Abstract

The world bank has classified 80 economies based on their Gross National Income (GNI) per capita as High-Income. European Medicines Agency (EMA), Food and Drug Administration (FDA), and Pharmaceuticals and Medical Devices Agency (PMDA) are the major regulatory stakeholders driving global pharmacovigilance regulations. The purpose of this article is to describe pharmacovigilance systems and processes in high-income countries, particularly those that are also members of the International Conference on Harmonization (ICH). All high-income countries are members of the WHO PIDM. The income level of a country has a direct relationship with medicine safety measures. All ten pioneering members of the Uppsala monitoring centre are from high-income countries and were the first responders after the thalidomide tragedy by making drug evaluation committees, introducing the ADR reporting forms and taking safety measures. Despite access to the VigiBase, some countries have separate databases for managing and analyzing data like Canada Vigilance online database, FDA Adverse Event Reporting System, the French pharmacovigilance database and European Union’s system Eudravigilance. All high-income countries have robust pharmacovigilance systems. USFDA and EMA are the world leaders in the field of pharmacovigilance. Most high-income countries follow EMA guidelines. Medicine safety is directly influenced by a country’s income level.

## 1. Introduction

Pharmacovigilance is highly regulated in the major regions of the world where medicines are developed. European Medicines Agency (EMA), Food and Drug Administration (FDA), and Pharmaceuticals and Medical Devices Agency (PMDA) are the major regulatory agencies driving global pharmacovigilance regulations. Legislation, regulations, guidance, and guidelines are established to define the department’s organizational structure, individual roles and systems, as well as to develop the skills necessary to perform pharmacovigilance efficiently. The Code of Federal Regulations in the USA, as well as national laws and ordinances in Europe, are legally binding [1]. 

Through collaboration between the World Health Organization, the Council for International Organizations of Medical Sciences (CIOMS) and the International Conference on Harmonization (ICH), pharmacovigilance has evolved into a regulatory activity [2]. There are three internationally recognized pharmacovigilance systems: the European Union (EU) pharmacovigilance system, the WHO Uppsala Monitoring Center system, and the International Conference on Harmonization (ICH) system. Pharmacovigilance characteristics vary across them, but they could all result in safe clinical drug use [3].

The approval of novel medications has been sped up, prioritized, and reviewed expeditiously in recent years [4]. As a result of the introduction of accelerated and conditional approval routes, new pharmacovigilance measures are required, as well as more frequent and creative risk management strategies. To meet the new challenges, FDA has taken additional measures [5]. 

It is hard to measure the impact of Pharmacovigilance related activities on public health. On knowing the ADRs, the regulatory actions of withdrawal or putting restrictions on various medicines leave unforeseen effects on public health. Several challenges lie ahead in the field of drug regulation and safety. A combination of advanced methodologies such as machine learning and the availability of large amounts of electronic healthcare data offers the potential for optimizing drug benefit-risk profiles in real-world environments. There have been increasing numbers of innovative therapeutics being developed and marketed lately, often via accelerated approval pathways, such as advanced therapy medicinal products, digital therapeutics, and vaccines developed using new technologies [6].

The world bank has classified 80 economies [7] based on their Gross National Income (GNI) per capita in current USD ($12,745 or more) [8] as high-income countries (Table 1). There was sufficient literature available on the pharmacovigilance systems in underdeveloped and developing countries [9,10,11,12,13,14]

Methodology and the literature included in this review. 

Only the literature related to high-income countries that have full membership of WHO PIDM was included in this study. The non-members or associate members of WHO PIDM were excluded from the study. Andorra, Barbados, Brunei Darussalam, Kuwait, Panama, the Republic of Korea, Oman, Singapore, Taiwan, and the United Arab Emirates were also excluded from the study due to the non-availability of sufficient information online in the English language. The pharmacovigilance systems of the European Union (EU) and European Economic Area (EAA) area member countries were not explained individually rather the overall pharmacovigilance system in the EU and EAA areas was investigated. The countries from which this information is synthesised and presented are listed in Table 2. The review focused on articles referring to selected high-income countries. A variety of sources were reviewed, including journal articles, websites, documents etc. The official websites of selected national regulatory authorities of high-income countries were reviewed between January 2022 to March 2022 for pharmacovigilance-related information. The information was also updated in November 2022. We also searched for relevant information on google. Between December 2021 and February 2022, relevant studies in English on pharmacovigilance related to the study aim were undertaken through a literature search on Google Scholar, Science Direct, PubMed, and Hinari to synthesize the extracted data. Publication dates were not limited. The following keywords were included “Pharmacovigilance” “ADR reporting system”, “Australia”, “Canada” “Chile”, “European Union”, “Israel”, “Japan”, “New Zealand”, “Saudi Arabia”, “United Kingdom”, “United States of America”, and “Uruguay”. The keywords were combined and incorporated into database searches.

To the best of our knowledge, no review has been conducted before on the pharmacovigilance systems in high-income countries. The purpose of this article is to describe pharmacovigilance systems and processes in high-income countries, particularly those that are also members of the ICH. 

## 2. Results

The pharmacovigilance system and the current practices in the selected high-income countries are discussed below. 

### 2.1. Pharmacovigilance System in Australia

The safety alert for thalidomide tragedy was first generated in Australia and this situation paved the path for the establishment of an Australian drug evaluation committee in 1963 by the national health department to investigate the safety of new medicines and a program for spontaneous reporting of ADRs. The Australian ADR reporting system was initiated with the sharing of reporting forms to prescribers in 1964. Australia is also one of the pioneer members of the WHO’s program for International Drug Monitoring. The adverse drug reaction advisory committee was formed in 1970 in response to the increasing number of reports. In 1971 “Blue card” form was introduced for clinicians to report. Now, these cards are not physically available and doctors can report any ADR via the Australian adverse drug reporting system [17,18].

Pharmacovigilance in Australia is largely dependent on voluntary reporting by patients and Healthcare Professionals (HCPs) and mandatory reporting by the industry. The HCPs can report medicine or vaccine adverse events online or by downloading a template. The downloaded template is installed in the best practice software for reporting. Consumers can report by telephone, online and by email. The industry is required to mandatory report ADRs through online, electronic data interchange and by email. The reports are stored in the Database of Adverse Event Notifications (DAEN). It includes information related to ADRs of prescription, over-the-counter, and complementary medicines and devices [15]. The Therapeutic Goods Administration publishes online information on medicine safety including medicines safety reviews, medicine safety guidelines, and scientific review reports. During the year 2020–2021, 57,771 medicine and vaccine adverse event reports were accepted by TGA. There is a 59% increase in reports than last year. The pharmaceutical industry submitted 14,418 (24%) during 2019–2020, 14,128 (61%) in 2020–2021 and HCPs contributed 4744 (20%) in 2019–2020, and 7960 (13%) in 2020–2021 of the total reports. Pharmacists were the toppers in sending AE (Adverse Event) reports among all HCPs [19]. 

The TGA is most interested in adverse events related to newly listed or registered drugs, medicines or vaccine interactions, not mentioned in literature or product information and events causing death or hospitalization. Following Europe, Australia has introduced the Black Triangle scheme to identify newly registered medicines and already registered drugs utilized in new ways requiring enhanced vigilance [16,17]. 

### 2.2. Pharmacovigilance System in Canada

Health Canada’s post-marketing surveillance program is called Canada Vigilance Program. This program was launched in 1965 to collect the suspected ADRs of the marketed products. The ADRs in the database can be searched online [20]. Like many other countries, there is a voluntary ADR reporting system for HCPs and consumers. Canadian Food and Drugs Act and regulations require market authorization holders (MAHs) to report all serious ADRs related to their marketed products. The MAHs also report serious ADRs occurring in foreign countries for products being marketed in Canada. The collected ADRs are stored in Canada Vigilance online database which can be accessed for information. There are seven Canadian regional vigilance centres which serve as contact points for HCPs and consumers and send information to Canada Vigilance National Office (centralized database) [21]. Since the beginning of the Canada vigilance program, the number of ADRs has increased annually. The contributing factors are an increase in the number of marketed products, mandatory reporting by hospitals, the industry’s patient safety programs, active surveillance programs etc. Health Canada received 96,559 domestic reports in 2019 while during the last 10 years, the number of reports has increased 4 times from 22,211 reports in 2010 to 96,559 reports in 2019. In one year 93.6% of reports were from mandatory reporters while during the last 10 years, 90% of reports were from industry [22]. 

Canada has shown its intent to amend the food and drug regulations and medical devices regulations in 2022. Among the various proposed amendments, there will be a requirement for a risk management plan from applicants for medicines and medical devices authorization [23]. 

### 2.3. Pharmacovigilance System in Chile 

In Chile, the concept of pharmacovigilance was first introduced in 1972 after a paper published by two researchers. The Chilean Institute of Public Health (ISP) launched the National Center for Drug and Pharmacovigilance (CENIMEF in Spanish, which is no longer official) in 1994 and developed a voluntary reporting system for suspected adverse reactions. As part of the ISP, a PV Committee was formed to analyze causality for adverse effects reported during the surveillance period, as well as an application to incorporate the centre into the WHO International Drug Surveillance Program. Chile was the fifth in South America to become a member of the Pharmacovigilance Program advocated by the World Health Organization (WHO) and the Uppsala Monitoring Center (UMC) [24,25].

In 2010, Chile published its first pharmacovigilance regulations, establishing its sub-department of pharmacovigilance (SDFV) within its regulatory agency (ISP) to coordinate pharmacovigilance activities. ISP considered the approach of the EMA and other regulators during the COVID pandemic and published a technical and regulatory guide, Implementation of Pharmacovigilance for SARS-CoV-2 vaccines in Chile, in December 2020. A stimulated passive surveillance system was also implemented by the regulator in addition to this framework, which involved sending virtual surveys to patients with COVID-19 vaccinations at three points in time: 48 h after each dose (inoculation), 7 days later, and 42 days afterwards [24].

In January 2016, hospitals in Chile’s capital, Santiago, and another regional hospital reported adverse effects linked to the use of metronidazole (an antibiotic commonly used to treat bacterial and parasitic infections of the skin, mouth, and genitals). A suspicious batch of metronidazole was withdrawn from sale by the manufacturer [26]. 

### 2.4. Pharmacovigilance System in European Union 

All EU member countries are full members of WHO PIDM [27]. After the wake of the shocking thalidomide catastrophe, to provide signals of unexpected adverse reactions, spontaneous adverse reaction reporting schemes have been developed. European Union enacted its first Community Directive (Council Directive 65/65/EEC) on medicines in 1965 [32]. A number of these directives provided a framework for the development of current medical systems, which are still in place today in the third millennium [35]. The overall objectives of regulatory pharmacovigilance include monitoring long-term safety in clinical practice to detect safety hazards that had not previously been identified or to determine if adverse effect profiles have changed, taking action to improve the safety of authorized medicines by assessing their risks and benefits, giving users information about their medicines so they can use them safely and effectively, and any action should be monitored to determine its impact [35]. In the early 1990s, EU member states began to cooperate more closely as proposals were made to build a more closely integrated regulatory system. Thus, the European Agency for Medication Product Evaluation was created in 1995. Later on, it was named as European Medicine Agency. Through cooperation with the European medicines regulatory network, EMA has achieved tremendous success—a partnership unique among the EMA, European Commission, and the medicines regulatory authorities in the European Economic Area [28,35]. 

The regulatory system in Europe is unique. The regulatory network is composed of the European Commission, the European Medicines Agency as well as the national competent authorities in member states of the European Economic Area (EEA). EMA’s work and success are based on the European medicines regulatory network. As well as coordinating and supporting interactions between 50 national competent authorities, the Agency also supports and coordinates efforts in veterinary medicine [28]. Scientists from across Europe participate in more than thirty EMA working groups to provide scientific expertise to regulatory processes. The regulatory pharmacovigilance in Europe has been enhanced two times in the new century. In 2004 the risk management approach was introduced through Regulation (EC) No. 726/2004 (EC, 2004) and in 2010 a new legislation was introduced in the EU area [35]. Eudravigilance was introduced by the EMA in 2001. Eudravigilance is the European Union’s system for managing and analyzing data related to suspected adverse reactions to medicines that have been approved for commercial use or are currently being studied in clinical trials. It enables the electronic transmission of individual case safety reports between all stakeholders, early identification and evaluation of potential safety signals; and product information in EEA. Since May 2004, Eudravigilance has included clinical trial data alongside case reports from around the world post-authorization. One of the EU countries. France also established a separate “French Pharmacovigilance data (FPVD)” in In 1985 [34].

This development conformed to the standards and formats of the International Council on Harmonization of Technical Requirements for Registration of Pharmaceuticals for Human Use [30,31,33]

The need for pharmacovigilance regulation in the EU arose when it was estimated that ADRs accounted for 5% of hospital admissions and 197,000 annual deaths. The European Commission reviewed the European system of safety monitoring and sponsored an independent study. Pharmacovigilance regulations were implemented in 2012 in the EU which was a revolutionary step in the field of medicine regulation [29]. The pharmacovigilance regulations implementations brought about transparency, stakeholders’ engagement, and safeguarding of public health. It also rationalized the responsibilities between the regulators and the pharmaceutical industry. 

European Union has integrated pharmacovigilance at all stages of the medical life cycle. The European Union has a separate data information system for suspected ADRs called Eudravigilance. EMA has launched this database for reporting suspected ADRs of authorized medicines and the medicines being studied during clinical trials for better analysis of the safety of medicines. The market authorization holders send reports to this database and don’t require sending them to NRAs. Similarly, EMA shares the information with WHO and NRAs do not need to send ICSR to vigilance. The patients and healthcare professionals send spontaneous ADR reports to NRAs. The reporting of unexpected ADRs during clinical trials will be shared with NRAs until the application of new clinical trial regulations. The public can also access the database through a separate portal [30]. 

700 million Euros was estimated as the cost of hospitalization due to avoidable ADRs in a UK-based study and the EU commission in 2008 calculated an EU-wide societal cost of ADRs 79 billion along with 197,000 deaths. EU adopted new pharmacovigilance legislation in 2010 which provides mandatory provisions to record ADRs reported by patients to MAHs and NRAs [33]. 

After a successful audit, the Eudravigilance database was declared fully functional and its system has met the functional specifications. On the recommendations of the pharmacovigilance Risk Assessment Committee (PRAC), an operational plan containing the key activities and developments for three years 2018–2020 was devised. EMA confirms that by the end of 2017, Eudravigilance contained a repository of more than 12.45 million ICSRs and 7.95 million cases along with information on 744,219 medicinal products. All ICSRs are also available to UMC [31]. 

### 2.5. Pharmacovigilance System in Israel 

Ministry of Health Israel created a Pharmacovigilance and Drug Information Department in 2012 within the Pharmaceutical Division. Israel has gone through a drug-related tragedy in 2011 commonly called a “levothyroxine event”. The drug was being marketed since 1981 and the company changed its composition. The adverse drug reactions associated with new formulations were noticed by many HCPs. Before this, there was no requirement for MAHs to report any ADRs to the Ministry of Health [36,37]. An investigation after the event revealed similar changes in formulations in Denmark in 2006 and New Zealand in 2007. Many adverse events were also reported as a result of these changes. The MAH was not able to transfer this vital PV information to the Israeli Ministry of Health on time, preventing the Ministry from taking preventive measures [37].

As a result of an amendment to the Pharmacists Regulations (Medical Products) 1986 in June 2013, all MAHs, and health maintenance organizations (HMOs) in Israel are now required to report ADRs and new safety information to the Ministry of Health. Swissmedic and the FDA are two leading medicine agencies with which Israel has signed agreements. Sharing of PV data is part of these memoranda of understanding. The total number of signals identified between 2014 and 2016 is 850 [37]. Israel’s Ministry of Health is taking a giant step toward international standards with its recent requirement that the pharmaceutical industry implements Risk Management Plans (RMPs) for product classifications that are considered high-risk [38].

### 2.6. Pharmacovigilance System in Japan

Japan kicked off the pharmacovigilance activities in 1967 from the selected medical institutions. In 1972 it became a member of WHO’s program for international drug monitoring (UMC) [27]. Japan established a post-marketing system in 1979 to assess the safety and efficacy of marketed medicines. Japan is considered the first country in the world which mandated pharmaceutical companies to require carrying out pharmacovigilance activities by establishing three systems i.e., the ADR collection and reporting system, the re-examination system, and the re-evaluation system. Pharmacies were included in this program in 1984 and from 1997 all pharmacies have joined the program. Good post-marketing surveillance system practice was introduced in 1993. In 2004 this program was divided under the revised pharmaceutical affairs law into good vigilance practice and good post-marketing study practice. Japan is considered one of the largest medicine markets in Asia. HCPs are legally required to report any suspected ADRs since 2003 [39].

### 2.7. Pharmacovigilance System in New Zealand

New Zealand started the pharmacovigilance program in 1965 and was among 10 countries that founded the WHO’s program for international drug monitoring. A separate unit named MedSafe is established in the Ministry of Health under DG Health to collect and review the safety reports of marketed products by the market authorization holders. The collection of suspected ADRs is contracted by the Ministry of Health to the centre for adverse reaction monitoring (CARM). The CARM is established in the Department of Preventive and Social Medicine at the University of Otago, Dunedin. Both MedSafe and CARM work closely on the reports and if need regulatory action is taken by the MedSafe. There is a Medicines Adverse Reactions Committee (MARC) to advise the Minister of Health on the safety of approved medicines [40,43].

The CARM contains a database of 110,000 reports which contains a large chunk from the HCPs. Pharmaceutical companies and patients also contribute to ADR reports significantly. In New Zealand, adverse reactions to medicines have been reported at the highest rate per capita for the last two decades [41].

Healthcare professionals can report ADRs online, with ADR reporting software, free post-yellow cards and a mobile application. There is an enhanced vigilance scheme for certain medicines to provide more information for investigating signal detection called M2 monitoring [42]. 

### 2.8. Pharmacovigilance System in Saudi Arabia

As an independent agency, the Saudi Food and Drug Authority (SFDA) was set up in 2003 to ensure the safety of foods, drugs, medical devices and biological substances. Within the drug, sector is the Vigilance and Benefit-Risk Assessment (VBRA) executive directorate. Since the NPC was launched in 2009, Saudi Arabia has been a full member of the WHO-UMC 2009, becoming the 92nd country to have done so. An EU GVP was adopted by the SFDA in 2015 (which was published in 2012), and an SFDA GVP was published that same year, becoming effective in 2016 [44]. The third version of the guidelines is published on 14 November 2022 [45]. A second Saudi Pharmacovigilance Guidelines on Good Pharmacovigilance Practices (GVP) was released in 2015 by the Saudi Food and Drug Authority [46,48]. SFDA safety alerts, guidelines on adverse drug event reporting by healthcare professionals and an information list on the safety of various brands are available on the website [47]. 

Healthcare providers and consumers/patients can submit online and paper forms, ADRs and quality defect reports, and the SFDA accept communications via the internet, mail, e-mail, fax, and phone. As part of its assessment of marketed medications’ safety, the VBRA evaluates ADRs and conducts data mining and signal detection based on these reports. There is an independent Pharmacovigilance committee that reviews drug profiles, comprising pharmacists, physicians, drug safety experts, and epidemiologists. In addition, since the Saudi GVP and the GVP for Arab countries are both based on the EU GVP, there are some minor variations, such as the ICSR submission requirements and the PSUR requirements for generics [44].

### 2.9. Pharmacovigilance System in the United Kingdom (UK)

After the thalidomide disaster, a committee on the safety of medicines (CSM) was established by the United Kingdom (UK). In 1965 yellow card scheme was introduced in the UK and the chairman of the CSM communicated to all HCPs especially doctors and dentists to report any unexpected reactions [53]. The suspected ADRs can be reported online or on the yellow card mobile application. Several safety alerts were generated from information received via yellow cards [49]. The pharmaceutical industry is legally required to report the ADRs [50]. A black triangle (▼) is displayed on the label and a summary of product characteristics of new medicines and vaccines that are under additional monitoring. All ADRs are reported for such products [51]. MHRA minimize the risk by taking regulatory actions in the form of product information changes Package label restricting the indications, changes from OTC to prescription medicines and publications in the drug safety updates [52]. 

### 2.10. Pharmacovigilance System in the USA

The US federal food, drugs and cosmetics Act was established in 1938 in aftermath of the sulfanilamide elixir (Diethyl glycol) tragedy which killed 107 people in 1937. The marketing authorization holders were required to demonstrate the safety of medicines before launching them into the market. The thalidomide tragedy in 1960 further creates an environment to strengthen the safety monitoring regulations. This time manufacturers were required to prove the efficacy of the medicines [54]. The USA was among the first ten pioneering members of the WHO’s PIDM in 1968. 

Usually, American patients are the first to receive the new drug molecules hence the chances of experiencing ADRs are also predominantly increased in the US population. The US pharmacovigilance system also faces the challenge of early detection of safety issues [56]. 

Among the FDA divisions, the Center for Drug Evaluation and Research (CDER) is responsible for approving and regulating drugs, the Center for Biologics Evaluation and Research (CBER) for biological products, the Center for Devices and Radiological Health (CDRH) for medical devices for human use. Office of surveillance and epidemiology works under CDER. FDA has 10 teams of safety evaluators’ mostly clinical pharmacists. The spontaneous ADR reporting by patients, consumers and healthcare professionals through MedWatch is voluntary. The MedWatch forms 3500A and 3500B are used for reporting ADRs by the HCPs, consumers and industry [55]. ADR reporting by the manufacturers is mandatory. The majority of the ADRs are from manufacturers which constitute 95% of the total reports received by the FDA [57]. All ADR reports are sent to the central database FDA Adverse Event Reporting System (FAERS). The system contains more than 9 million reports since 1969. The FDA’s CFR title 21 contains various sections including the ones that encompass the safety requirements for medicines and medical devices for human use. The 21 CFR, 314.80 requires that post-marketing safety reports must be submitted to the FDA within 15 days from all domestic and foreign sources and periodic ADEs reports as per schedule. When any safety concern comes to the surface by FAERS then further evaluation is done extending to more studies using the large databases. Any such record is maintained for ten years [58]. Vaccine Adverse Event Reporting System (VAERS) is separately provided for the public to report ADEs/AEFI related to vaccines (https://vaers.hhs.gov/reportevent.html, accessed on 22 July 2022) [59]. WONDER, an online database is utilized for inquiries regarding vaccine adverse events. A number of useful guidelines for the industry have been provided including E2E Pharmacovigilance Planning Guidance, Post-marketing Studies and Clinical Trials Guidance, and good pharmacovigilance practices [59]. US follows the ICH guidelines being the founding member. The US does not require a qualified person for the pharmacovigilance and pharmacovigilance system master file [50].

### 2.11. Pharmacovigilance System in Uruguay

The pharmacovigilance activities were started in Uruguay in the year 2000 [62]. The Pharmacovigilance Unit was created in 2006 in the ministry of Health Uruguay. By adopting Good Pharmacovigilance Practices for the Americas (WHO), this Unit collaborates with its National Advisory Committee, formed the same year, and provides technical assistance. Members include members of the university (Faculty of Medicine, Faculty of Chemistry), members of the public health system, and others. Uruguay became a member of the WHO PIDM in 2001 [61]. Pharmacovigilance is included in the course of Pharmaceutical Care and it is also incorporated in some post-graduate courses.

As a result of Ministerial Ordinance No. 292, the additional surveillance modality has been added to the National Pharmacovigilance System as an intermediate step between passive (spontaneous reports) and active (intensive) pharmacovigilance. 12 December 2014, Ordinance No. 798. Additional surveillance will apply to medicines that contain a new active ingredient, biotechnological medicines, or those for which data regarding post-authorization is required. The Department of Medicine will define which medicines are covered by this additional surveillance modality [60]. Uruguay has given a priority to patient safety in dentistry as well and established an online ADR reporting system for dental products [62].

## 3. Discussion

All 80 high-income countries are members of the WHO PIDM. There is a cost to maintain the PV activities in the country. The budgetary constraints are one of the reasons for non-functional PV systems in LMIC. The high-income countries have the potential to spend on PV activities that’s why have functional PV systems. The income level of a country has a direct relationship with medicine safety measures.

The ten pioneering members of the Uppsala monitoring centre i.e., Australia, Canada, Federal Republic of Germany, Ireland, Netherlands, New Zealand, Sweden, United Kingdom, USA and former Czechoslovakia [63] are high-income countries. After Slovak tensions, Czechoslovakia peacefully divided into the Czech Republic and Slovakia and both are also high-income countries. These countries were the first responders after the thalidomide tragedy by making drug evaluation committees, introducing the ADR reporting forms and taking safety measures.

VigiBase is a global PV database and all high-income countries have access to that system. Some high-income countries have established separate databases for managing and analyzing information on suspected adverse drug reactions. These databases include the Australian Database of Adverse Event Notifications “DAEN”, European Unions’ Eudravigilance, the Canada Vigilance online database, the FDA Adverse Event Reporting System, and the French pharmacovigilance database.

One of the most sophisticated and complete pharmacovigilance systems in the world, the EU system serves as a reliable and open tool to guarantee a high degree of public health protection across the EU. Many high-income countries follow the EU guidelines. The European Union (EU) PV system is among three internationally recognized PV systems. EU introduced the risk management approach first time in 2004. Australia is following Europe in introducing the Black Triangle scheme, While Saudi Arabia has adopted the EU GVP and Chile tracks EMA during the COVID pandemic. Being the founding member US follows the ICH guidelines. Israel has signed MoU with Swissmedic and USFDA for safety data sharing. No other country shares safety reports to other countries other than VigiBase. By establishing three systems—the ADR collecting and reporting system, the re-examination system, and the re-evaluation system—Japan is regarded as the first nation in the world to oblige pharmaceutical companies to carry out pharmacovigilance activities.

## 4. Conclusions

All high-income countries have robust pharmacovigilance systems. USFDA and EMA are the world leaders in the field of pharmacovigilance. Most high-income countries follow EMA guidelines. The income level of a country has a direct impact on medicine safety. The robustness of Australia, the EU, France, the US, and Canada is proven by their independent PV databases.

## Figures and Tables

**Table 1 pharmacy-11-00010-t001:** List of High-Income Countries.

WHO PIDM Non-Member			
1. Aruba,	9. Gibraltar,	17. Nauru,	25. St. Kitts and Nevis,
2. Bermuda,	10. Greenland,	18. New Caledonia,	26. St. Martin (French part),
3. Cayman Islands,	11. Guam,	19. Northern Mariana Islands,	27. Taiwan China,
4. Channel Islands,	12. Hong Kong SAR China,	20. Puerto Rico,	28. Trinidad and Tobago,
5. Curaçao,	13. Isle of Man,	21. Seychelles,	29. Turks and Caicos Islands
6. Czech Republic,	14. Liechtenstein,	22. San Marino,	30. Virgin Islands (U.S.)
7. Faroe Islands,	15. Macao SAR China,	23. Sint Maarten (Dutch part),	
8. French Polynesia,	16. Monaco,	24. Slovak Republic,	
WHO PIDM Associate members			
31. Bahamas,	32. Antigua and Barbuda,	33. Bahrain	34. Qatar
European Union (EU) and European Economic Area member countries			
35. Austria,	42. France,	49. Latvia,	56. Portugal,
36. Belgium,	43. Germany,	50. Lithuania,	57. Romania,
37. Croatia,	44. Greece,	51. Luxembourg,	58. Slovenia,
38. Cyprus,	45. Hungary,	52. Norway (EAA)	59. Spain,
39. Denmark,	46. Iceland (EAA)	53. Malta,	60. Switzerland
40. Estonia,	47. Ireland,	54. Netherlands,	61. Sweden (EAA)
41. Finland,	48. Italy,	55. Poland,	
Other WHO PIDM full-member countries			
62. Andorra,	67. Canada	72. Oman	77. United Kingdom of Great Britain and Northern Ireland
63. Australia	68. Israel	73. Panama	78. United States of America
64. Barbados	69. Japan	74. Republic of Korea	79. United Arab Emirates
65. Brunei Darussalam	70. Kuwait	75. Saudi Arabia	80. Uruguay
66. Chile	71. New Zealand	76. Singapore	

**Table 2 pharmacy-11-00010-t002:** Additional Comments.

S. No	Name of the Participating Country	Sources of Information
1.	Australia	Website [15,16]
		Peer Review paper [17]
		Book [18]
		Reports [19]
2.	Canada	Website [20,21,22,23]
3.	Chile	Peer Review paper [24]
		Book [25]
		Report [26]
4.	European Union	Website [27,28,29,30,31]
		Peer Review papers [32,33,34]
		Book [35]
5.	Israel	Website [36]
		Peer Review papers [37,38]
6.	Japan	Peer Review papers [39]
7.	New Zealand	Website [40,41,42]
		Peer Review paper [43]
8.	Saudi Arabia	Website [27,44,45,46,47]
		Peer Review paper [48]
9.	United Kingdom	Website [49,50,51,52]
		Book [53]
10.	United States of America	Website [50,54,55]
		Peer Review papers [56,57,58,59]
11.	Uruguay	Website [60]
		Peer Review paper [61,62]

## Data Availability

Not applicable.

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
