# Peer review of "Pharmacovigilance in High-Income Countries: Current Developments and a Review of Literature"

_pharmacy, 2023, doi:10.3390/pharmacy11010010_

Round 1

Reviewer 1 Report

I believe that the topic of this manuscript is very relevant. Indeed, review of the pharmacovigilance systems in high-income countries has not been studies previously systematically nor have pharmacovigilance systems and processes in high-income countries been described in depth.

However, the study is poorly designed.

First, too short of a time period during which the literature was reviewed was taken.

The authors did not have any objective criteria by which to evaluate different pharmacovigilance systems, they merely described them and listed whatever information they found. They did not describe the same features for all systems, for example, whether patient registration is enabled, how they approve and inform about risk minimization measures..etc.

For example, it would have been good to explore the following issues: is there a reporting system for a specific country and if yes, how it functions, the number of reports per 10,000 inhabitants, the quality of the report, the number of signals that have been identified, do they conduct epidemiological studies to measure and characterize the risk, do they measure the effect of risk minimization measures, do they publish risk minimization measures on their website - what is their risk communication to the public.

Author Response

Dear Reviewer No. 1

We are thankful to you for your valuable comments. The study design is as follows that has been added at the end of the introduction. It was not previously mentioned as per journal’s format.  This review is not a systematic review of literature hence risk of bias criteria would not apply on this paper. Due to the scarcity of original research on pharmacovigilance in high-income countries, websites have also been used as a resource.

Furthermore, it is a review hence we could only present the information that have been found in literature. Having said so we have presented the information related to functioning of Australian PV system at lines 151-167. Similarly, the information how Canadian PV system works is described at lines 183-185. 

The following has been included in the introduction section.

Only the literature related to high-income countries who have full membership of WHO PIDM were included in this study. The non-members or associate members of WHO PIDM were excluded from the study. Andorra, Barbados, Brunei Darussalam, Kuwait, Panama, Republic of Korea, Oman, Singapore, Taiwan, and United Arab Emirates were also excluded from study due to non-availability of sufficient information online in English language. The pharmacovigilance systems of European Union (EU) and European Economic Area (EAA) area member countries were not explained individually rather the overall pharmacovigilance system in the EU and EAA areas was investigated. The countries from which this synthesis and information is presented are listed in Table 2.  The review focused on articles referring to selected high-income countries. A variety of sources were reviewed, including journal articles, websites, documents etc. The official websites of selected national regulatory authorities of high-income countries were reviewed between January 2022 to March 2022 for pharmacovigilance-related information. The information was also updated in November 2022. We also searched for relevant information on google. Between December 2021 and February 2022, relevant studies in English on pharmacovigilance related to the study aim were undertaken through a literature search on Google Scholar, Science Direct, PubMed, and Hinari to synthesize the extracted data. Publication dates were not limited. The following keywords were included “Pharmacovigilance” “ADR reporting system”, “Australia”, “Canada” “Chile”, “European Union”, “Israel”, “Japan”, “New Zealand”, “Saudi Arabia”, “United Kingdom”, “United States of America”, and “Uruguay”. The keywords were combined and incorporated into database searches.

Reviewer 2 Report

The current manuscript addresses the pharmacovigilance systems throughout the world. It is quite interesting since it gives a very broad perspective on the characteristics of pharmacovigilance in various countries. Nevertheless, some alterations should be made:

- The abstract should contain more results, discussion and conclusions, since the grand majority of it regards the introduction and methods section only;

- There should be a final discussion section mentioning the aspects of the pharmacovigilance systems of each country that are similar, and also what makes them different;

- The conclusion section should also be extended, to include these comparison discussion between the countries’ pharmacovigilance systems (differences and similarities).

Author Response

Dear Reviewer No. 2

We are grateful to you for suggestions to improve the manuscript. As advised the abstract has been revised. The final discussion and conclusion sections have also been revised as per advice.

The pharmacovigilance systems of the high-income countries are already developed. However, there is scarcity of literature on the topic. Hence, we have used other resources to extract the information. This also includes drug regulatory authorities’ websites.  

Regards, 

Reviewer 3 Report

Important work presenting the state of pharmacovigilance in high income countries. The content is legible and understandable.

My comments include references issues. I am not sure if item 3 is justified. Apart from the abstract, the work is in Chinese. The work is about China. In addition, please pay attention to items 18, 26, 32, 36. Please give more details of access, if online I propose to add a links. Item 58 please specify access details. I can not find them. Item 59 for Uruguay is from 2014, is there more recent literature available.

Interesting and informative paper.

Author Response

Dear Reviewer No. 3

We highly appreciate your comments on our manuscript. The reply to the comments is as under.

The information related to three internationally recognized pharmacovigilance systems presented in the introduction is extracted from abstract of the Chinese paper and cited at reference 3.

The references at serial 18, 26, 32 and 36 have been reviewed and corrected where required. The reference at Sr. 58 has been added with a link. The relevant literature on Uruguay is mostly in Spanish. Also, one more reference is added at Sr. No. 62. 

Regards,